# Impacts of Selenium–Chitosan Treatment on Color of “Red Globe” Grapes during Low-Temperature Storage

**DOI:** 10.3390/foods13030499

**Published:** 2024-02-04

**Authors:** Wei Wang, Yaping Liu, Jianbing Di, Yu Wang, Bing Deng, Jiali Yang, Zezhen Li, Lixin Zhang

**Affiliations:** College of Food Science and Engineering, Shanxi Agricultural University, Jinzhong 030800, China; aweiaway23@163.com (W.W.); dijianbing@126.com (J.D.); sxtgwy@126.com (Y.W.); dengbing@sxau.edu.cn (B.D.); jiayang@sxau.edu.cn (J.Y.); sxnydxzezhen@sxau.edu.cn (Z.L.); 13934600160@139.com (L.Z.)

**Keywords:** selenium, chitosan, grape storage, skin color, anthocyanin, gene expression

## Abstract

Maintaining the vibrant color of fruit is a longstanding challenge in fruit and vegetable preservation. Chitosan and selenium, known for their protective and antioxidant properties, have been applied to preserve these produce. This study aimed to investigate the influence of selenium–chitosan treatment (comprising 25 mg L^−1^ selenium and 1.0% chitosan) on the color of “Red Globe” grapes and to analyze the relative expression of genes associated with anthocyanin synthesis enzymes (*VvCHS*, *VvCHI*, *VvF3H*, *VvF3′H*, *VvF3′5′H*, *VvDFR*, *VvLDOX*, *VvUFGT*, *VvOMT*, *Vv5GT*, and *VvGST*) using RT-qPCR. Our goal was to uncover the regulatory mechanisms governing grape color. Comparing various treatments, we observed that selenium–chitosan treatment had a significant effect in reducing decay, maintaining the soluble solids content of grape flesh, and preserving the vivid color of grape. This research indicated that selenium–chitosan treatment slowed down browning and prevented the reduction in total phenolic, flavonoids, and anthocyanin in the grape. Moreover, gene expression analysis revealed that selenium–chitosan treatment increased the expression of *VvCHS*, *VvF3H*, *VvF’3′H*, *VvLDOX*, and *Vv5GT*, while also stabilized the expression of *VvCHI*, *VvF3′H*, and *VvDFR* in grape skins. These findings shed light on the potential mechanism by which selenium–chitosan impacts grape color. This study established a theoretical foundation for investigating the molecular mechanisms behind selenium–chitosan’s ability to slow down grape browning and provides a novel approach to enhancing fruit and vegetable preservation techniques.

## 1. Introduction

Grapes (*Vitis vinifera*) stand as a cornerstone among globally cherished and frequently consumed fruits, holding an important role in human life as both a pivotal cash crop and a source of nourishment [1,2]. Besides their fundamental importance, grapes demonstrate an extensive range of applications and potential across diverse domains, including agriculture, food science, nutrition, and beyond. The “Red Globe” grape variety typically boasts a rich deep red or purplish color [3] and is abundant in anthocyanins, vitamins, and minerals known for their positive effects on human health [4,5,6]. However, during storage and transportation, the skins of “Red Globe” grapes are particularly vulnerable to browning, a physiological anomaly believed to arise from the high content of phenolic and polyphenolic compounds. When these compounds interact with the oxygen in the air, they incite oxidative reactions, resulting in the gradual darkening of skin color and the emergence of undesirable browning attributes. This phenomenon not only diminishes the commercial value of the produce but also inflicts substantial losses on both growers and distributors. Several studies of changes in grape skin color and chemical composition have emphasized the key role of anthocyanins [7]. Anthocyanins, a group of naturally occurring plant compounds, are abundant in various fruits and vegetables, with grapes containing significant concentrations. These compounds give grapes their captivating purple, red, and blue color, thereby significantly affecting their visual appeal and coloration [8]. Furthermore, anthocyanins bring forth excellent antioxidant properties, rendering them advantageous for human health. The intricate pathway of anthocyanin biosynthesis has been extensively elucidated, and the majority of genes governing anthocyanin biosynthesis and regulation have been isolated and scrutinized across various model plants [9]. In grape skin, anthocyanins are synthesized via the flavonoid pathway [10]. Scientists have identified that anthocyanin synthesis and accumulation closely intertwine with the expression levels of a cluster of genes that encode enzymes involved in the anthocyanin biosynthetic pathway. These encompass chalcone synthase (CHS), chalcone isomerase (CHI), flavanone 3-hydroxylase (F3H), flavonoid 3′-hydroxylase (F3′H), flavonoid 3′,5′-hydroxylase (F3′5′H), dihydroflavonol 4-reductase (DFR), leucoanthocyanidin dioxygenase (LDOX), UDP-glucose: flavonoid 3-O-glucosyltransferase (UFGT), O-methyltransferase (OMT), UDP-glucose: flavonoid 5-O-glucosyltransferase (5GT), and glutathione S-transferase (GST), among others [9].

Selenium, a trace element present in minute quantities within the human body, holds undeniable importance for health. It plays a crucial role in bolstering the immune system, while offering a host of advantageous properties such as antioxidant properties, anti-aging effects, and the potential for cancer prevention [11,12,13]. Selenium also serves as an integral component in various enzymes, notably selenoproteins like glutathione peroxidase [14]. These enzymes actively engage in antioxidant reactions within the tissues of fruits and vegetables, diligently combating free radicals and oxidizing substances, thus effectively retarding the oxidation process [13]. Beyond its antioxidant capability, selenium possesses antimicrobial properties [15], effectively inhibiting the proliferation of bacteria and fungi. Moreover, it contributes to fortifying cell membranes, diminishing their permeability [16]. Sodium selenite, a selenium compound, can affect the activity of specific enzymes in fruits and vegetables, such as polyphenol oxidase [17], consequently defeating food aging and spoilage by impeding naturally occurring enzyme-facilitated biochemical reactions. Additionally, selenium ions in sodium selenite can form complexes with metal ions present in foods [18], aiding in curbing unstable oxidation or reduction reactions involving metallic ions and pigment molecules. Although many studies have emphasized the potential of selenium treatments in maintaining the quality of stored fruits and vegetables [13,19], their application in the food industry encounters constraints related to concerns about their toxicity and limited bioavailability [20].

Chitosan, a biopolymer compound derived from natural resources such as shells and insects, emerges through the deacetylation process of chitin under alkaline conditions [21]. Its exceptional biocompatibility, degradability, and antimicrobial attributes render chitosan a prevalent choice in the fields of food packaging, preservation, and freshness conservation [22,23,24]. Chitosan coating treatments create a protective physical barrier on the surfaces of fruits and vegetables. This shield effectively prevents water evaporation, sustains the structural integrity of cells, and mitigates skin discoloration and moisture loss induced by evaporation [25]. Furthermore, chitosan has been explored for its utility as an antibacterial coating [26]. By modulating the physiological and metabolic processes of fruits and vegetables, such as decreasing respiration rate, diminishing ethylene production, and inhibiting enzymatic activities, chitosan coating treatments effectively decelerate the ripening and decay of produce [24]. In addition to its fruit and vegetable applications, chitosan has been widely used in the preservation of meat, eggs, and aquatic products [27,28,29]. Nevertheless, to rectify the inherent constraints of single chitosan coating, they are typically incorporated into composite membranes alongside other constituents, such as inorganic materials [30] or plant extracts [31], to fulfill a broader spectrum of food preservation needs.

Studies have shown that the combination of selenium and polysaccharides can overcome these deficiencies. Chitosan, abundant in amino and hydroxyl functional groups, exhibits diverse biological activities [32]. Currently, the function of selenium–chitosan has been investigated in animal studies. For instance, CTS-SeNPs exhibit high antidiabetic activity [33], while LCS-SeNPs hold promise for treating oxidative-stress-related inflammatory bowel diseases [34]. However, limited attention has been given to the application of selenium–chitosan in the field of fruit and vegetable preservation. Notably, selenium–chitosan treatment has been demonstrated to extend the storage lifespan of grapes in our previous study [35,36]. And, during our experiments, we found color changes in grapes treated with selenium–chitosan but the mechanism is not clear yet. Our group also demonstrated that this treatment effectively inhibits the yellowing of fresh-cut broccoli, delays chlorophyll degradation and carotenoid synthesis, and suppresses the expression of transcription factors associated with HYD and CAO [37]. Therefore, on the basis of these experiments, there is a great need to investigate the mechanism of selenium–chitosan on the color of grapes during storage. Given the synergistic potential of chitosan and selenium in fruit and vegetable preservation, the present study focused on investigating the impacts of selenium–chitosan composite coating on the skin color of harvested grapes. By studying the shifts in the expression of genes implicated in the anthocyanin synthesis pathway, this research aimed to unveil the molecular mechanism of grape skin color transformation. The findings from the current study were expected to lay the theoretical basis for the application of selenium–chitosan in the conservation and retention of color in fruits and vegetables.

## 2. Materials and Methods

### 2.1. Plant Material and Treatments

“Red Globe” grapes were harvested from Shanxi Agricultural University Pomology Institute. All grapes had a soluble solid content greater than 16%, indicating that they were fully ripe. They underwent meticulous selection, with only undamaged, disease-free, and uniformly sized, ripened, and colored grapes qualifying for subsequent testing.

Based on the results of past experiments conducted by our group, it was determined that the optimal treatment for the grapes involved a 2 min immersion in a solution comprising 25 mg L^−1^ sodium selenite and 1.0% chitosan [35,36].

The selenium–chitosan solution was prepared by adding 1.0% chitosan and 25 mg L^−1^ sodium selenite into distilled water. While agitating the mixture, 1.0% glacial acetic acid was slowly added until the solution achieved clarity. The grapes were divided into small clusters and subjected to four distinct treatments: immersion in distilled water (CK), a sodium selenite solution (25 mg L^−1^, Se), a chitosan solution (1.0%, CS), and a selenium–chitosan solution (25 mg L^−1^ selenium and 1.0% chitosan, CS+Se), each for a duration of 2 min. Subsequently, the grapes were blown dry and stored in 0.03 mm polyethylene bags at 0 °C in a cold storage facility. All experiments were repeated three times, except where noted.

### 2.2. Decay Rate

Grapes that are visibly wrinkled or covered with mold are considered to be rotten. All grapes in storage were treated as statistical objects. The weight of decayed grapes was recorded at 15-day intervals, and the decay rate of the grapes was determined using the following formula:(1)$$\begin{array}{c} Decay\;rate\left(\%\right)=\frac{weight\;of\;decayed\;grapes}{total\;weight\;of\;grapes}\times 100 \end{array}$$

### 2.3. Soluble Solids Content

Soluble solids content is measured using a pocket refractometer (ATAGO, Tokyo, Japan). The pulp of the grapes was wrapped in gauze, the juice was squeezed out, and drops were read on the pocket refractometer.

### 2.4. Chromatic Aberration

To assess the degree of browning, the L*, a*, and b* values of grape skins were measured using a handheld colorimeter (YS3060, 3nh Technology, Shenzhen, China). Ten grapes were taken from each group and each of the four equatorial sides of the grapes were measured using a handheld colorimeter.
(2)$$\begin{array}{c} \Delta E=\sqrt{(\Delta {L)}^{2}+(\Delta {a)}^{2}+(\Delta {b)}^{2}} \end{array}$$

### 2.5. Total Phenolic, Flavonoid, and Anthocyanin Content

A total of 0.5 g of pre-crushed pericarp tissue powder was weighed and mixed with a small amount of pre-chilled 1.0% HCl–methanol solution. The mixture underwent homogenization within the confines of an ice bath and was subsequently transferred to a 20-milliliter graduated test tube. To ensure no loss of the sample, the mortar was rinsed with 1.0% HCl–methanol solution, and the rinse was then added to the test tube. The volume of the solution was adjusted to match the scale, after which it was thoroughly mixed. The resulting blend was subjected to an extraction process in the absence of light at 4 °C for a duration of 20 min, during which the tubes were shaken with a vortex shaker (Kylin-Bell, Haimen, China) every 2 min. Subsequently, the mixture was filtered and the resultant filtrate was gathered for further analysis. The 1.0% HCl–methanol solution was employed as a reference for zero adjustment. The absorbance values of the solutions were then measured at specific wavelengths, namely 280 nm, 325 nm, 600 nm, and 530 nm, using a UV spectrophotometer (Cary 60, Agilent Technologies, Santa Clara, CA, USA). To determine the total phenolic content, a standard curve was established using various concentrations of gallic acid, with the equation y = 1.34182x + 0.01945 and an R^2^ value of 0.99777, where “x” referred to the milligram equivalent of gallic acid and “y” represented the absorbance value of the solution at 280 nm. Another standard curve, employing various concentrations of rutin, was used to measure the flavonoid content, with the equation y = 9.1725x − 0.01062 and an R^2^ value of 0.99543, where “x” represented the milligram equivalent of rutin and “y” represented the solution’s absorbance value at 325 nm. The anthocyanin content (U) was calculated as the variation between the absorbance values at 530 nm and 600 nm, following the formula U = (OD_530_ − OD_600_) [38].

### 2.6. Quantitative Real-Time PCR (RT-qPCR) Assay

The RNAprep Pure Plant Plus Kit (polysaccharides- and polyphenolics-rich) (DP441, Tiangen Biotech, Beijing, China) was used for the comprehensive extraction of total RNA from grape skins. To assess RNA concentration and purity, the extracted samples were analyzed using an ultra-micro UV spectrophotometer (Nanodrop One C, Thermo Fisher Scientific, Waltham, MA, USA). The TransScript^®^ Uni All-in-One First-Strand cDNA Synthesis SuperMix for qPCR (One-Step gDNA Removal) kit (AU341, TransGen Biotech, Beijing, China) was subsequently employed to convert the total RNA of each sample into first-strand cDNA. The cDNA was quantified using a fluorescent quantitative PCR instrument (BIO-RAD, Hercules, CA, USA) in conjunction with the PerfectStart^®^ Green qPCR SuperMix kit (AQ601, TransGen Biotech, Beijing, China). The RT-qPCR reaction mixture comprised a total volume of 20 μL, including cDNA (2 μL), each primer 0.4 μL, 10 mM (Sangon Biotech, Shanghai, China), nuclease-free water (7.2 μL), and 2× PerfectStart^®^ Green qPCR SuperMix (10 μL). The thermal cycling parameters involved an initial denaturation step at 94 °C for 30 s, followed by 40 cycles of denaturation at 94 °C for 5 s and extension at 60 °C for 30 s. For reference, the primer sequences for the target genes are listed in Table 1. Each experiment was replicated three times. The expression levels of the target genes were determined using the 2^−ΔΔCT^ method [39].

### 2.7. Statistical Analysis

Statistical analysis encompassing linear regression and correlation analyses were carried out with Origin Pro 2022 (Origin Lab Inc., Northampton, MA, USA). For computing standard deviations, one-way ANOVA tests, and further correlation analyses, SPSS 26 (SPSS Inc., Chicago, IL, USA) served as the analytical platform. Cluster analysis was performed using R Studio 2023.06.1 (Posit Software, Boston, MA, USA).

## 3. Results

### 3.1. Decay Rate and Soluble Solids Content

The decay rate is a significant parameter for the assessment of grape quality throughout the storage period. As illustrated in Figure 1A, the decay rate of grapes progressively increased across all groups as the storage duration advanced. By the 30th day, decay became apparent in all grape groups, with the CK group showing a more pronounced decay rate compared to the other treatment groups. By the 45th day, a notable trend emerged, revealing that the grape decay rate in the CS+Se group was lower than that in the CK, Se, and CS groups. While both Se and CS treatments contributed to a reduction in the rate of grape decay, the combined CS+Se treatment demonstrated a more pronounced effect.

Soluble solids content can reflect the ripeness and storage quality of the grapes. As shown in Figure 1B, the soluble solids content of the CS+Se treatment group recovered to the initial level after a brief decline at the beginning of storage, which could be attributed to the loss of water, while the soluble solids content of the other treatment groups continued to decline and the difference between the groups was significant (*p* < 0.05) by the 60th day.

### 3.2. Chromatic Aberration

To understand the chromatic aberration of grape skin color, it is essential to consider the L*, a*, b*, and ΔE values. L*, a*, and b* values were used to determine brightness, red–green, and yellow–blue color, respectively. ΔE, indicating the degree of browning, reflected alterations in grape skin color, with higher ΔE values indicating more pronounced browning. Across all groups, the L* (Figure 2A) and a* (Figure 2B) values exhibited a declining trend during the storage period. The CK group displayed the most rapid decline in both L* and a* value, while the CS+Se group showed a notably slower decrease. The b* values (Figure 2C) in each group exhibited a gradual increase with storage time; yet, on the 60th day, a sudden and substantial upsurge in the b* value was observed in the CK group, whereas insignificant changes were noted in the other groups. The ΔE value (Figure 2D) was increased most rapidly in the CK group but exhibited the slowest increase in the CS+Se group. By the 60th day, the ΔE value in the CS+Se group was only 58.06% of that in the CK group, and the CS+Se treatment significantly hindered the escalation of ΔE value (*p* < 0.05). These findings suggested that both selenium and chitosan were effective in preserving the natural color of grape skins, with the most remarkable outcomes observed in the CS+Se group. The different treatment effects on grape phenotypes were recorded (Figure 2E). The results indicated that Se, CS, and CS+Se treatments all inhibited grape decay and suppressed grape browning, with CS and CS+Se treatments exhibiting the most effective outcomes.

### 3.3. Analysis of Total Phenolic, Flavonoid, and Anthocyanin Content

A comprehensive evaluation of the content of total phenolic, flavonoid, and anthocyanin within grape skins was conducted. The trends observed in the total phenolic (Figure 3A), flavonoid (Figure 3B), and anthocyanin (Figure 3C) content of each group followed a pattern of initial increase followed by a subsequent decrease during the storage period. Throughout the storage duration, the total phenolic content within all three treatment groups was higher than that of the CK group, particularly in the CS+Se group, where the total phenolic content soared to 55.32% higher than that of the CK group by the 60th day. The flavonoid content in the CK group displayed an initial increase until the 30th day, exceeding both the Se and CS groups. Subsequently, it sharply declined, significantly plummeting below the levels observed in other groups by the end of the storage period (*p* < 0.05). Overall, the CS+Se group exhibited higher flavonoid content compared to the other groups. However, no statistically significant difference was observed between the CS+Se group and the Se or CS groups. In contrast, the anthocyanin content in the CK group exhibited a continuous decreasing trend, plummeting to only 32.44% of that in the CS+Se group by the 60th day. The CK group experienced a substantial loss of pericarp pigmentation due to microbial infestation during the latter stages of storage. Overall, the CS+Se demonstrated advantages in preserving total phenolic, flavonoid, and anthocyanin content. Notably, the chitosan treatment appeared to exert a greater impact on maintaining anthocyanin content compared to the selenium treatment.

### 3.4. Analysis of Gene Expression of Anthocyanin-Synthesis-Related Enzymes

An assessment was conducted to determine the relative expression levels of various genes associated with anthocyanin synthesis in grape skins, encompassing *VvCHS*, *VvCHI*, *VvF3H*, *VvF3′H*, *VvF3′5′H*, *VvDFR*, *VvLDOX*, *VvUFGT*, *VvOMT*, *Vv5GT*, and *VvGST*. The relative expression of these genes was analyzed using RT-qPCR under different treatment conditions across different storage periods.

*VvCHS*, which encodes chalcone synthase and serves as the initiation point in the flavonoid pathway, displayed relatively stable expression trends in the CK and Se groups, exhibiting no obvious dramatic alterations. Conversely, in the CS and CS+Se groups, *VvCHS* expression substantially increased after 30 days of storage, showing a distinct trend compared to the CK and Se groups (Figure 4A). *VvCHI* is responsible for encoding chalcone isomerase. The expression patterns of *VvCHI* demonstrated variations among the different treatment groups. In the CS and CS+Se groups, *VvCHI* expression remained relatively constant during storage. In contrast, the CK and Se groups exhibited a significant increase in *VvCHI* expression during the latter stage of storage with fluctuations. This fluctuation may be correlated with the occurrence of decay in the CK and Se groups at the 30th and 45th day, respectively, suggesting that decay potentially triggered alterations in cellular metabolism. Interestingly, both the CS and CS+Se groups showed elevated *VvCHI* expression by the 60th day, coinciding with grape rotting, indicating a potential link between the elevated expression of *VvCHI* and the process of grape rotting (Figure 4B).

*VvF3H*, responsible for encoding flavanone 3-hydroxylase, displayed distinctive trends in the CS+Se group at the beginning of storage, exhibiting an opposite pattern to the other groups by suppressing *VvF3H* expression. The CK group showed a continuous decline in *VvF3H* expression, with levels lower than those observed in the other treatment groups. In contrast, the Se group demonstrated a sudden increase in *VvF3H* expression on the 60th day (Figure 5A). The overall expression of *VvF3H* during the middle and late stages of storage appeared to be intricately influenced by the different treatments. The expression of *VvF3′H*, which is responsible for encoding flavonoid 3′-hydroxylase, displayed a similar pattern of changes across all groups, characterized by fluctuations followed by stabilization (Figure 5B). During the pre-storage period, all treatment groups exhibited an upregulation in the expression of *VvF3′H*, with the CS group being the most significant (*p* < 0.05). Regarding *VvF3′5′H*, responsible for encoding flavonoid 3′,5′-hydroxylase, the expression pattern of *VvF3′5′H* exhibited inconsistent fluctuations across groups during storage. Nevertheless, on the whole, all treatment groups showed an upregulation in *VvF3′5′H* expression (Figure 5C).

*VvDFR*, responsible for encoding dihydroflavonol-4-reductase, demonstrated varying expression patterns across different treatment groups. In the CK group, *VvDFR* expression exhibited an increase followed by a decrease over the storage duration. The CS+Se group initially suppressed the upregulation of *VvDFR* expression at the beginning of storage but showed an upsurge in *VvDFR* expression as storage progressed (Figure 6A). This suggested that the CS+Se treatment initially had a negative regulatory impact on *VvDFR* expression during early storage but decreased in the regulation during the later stage.

*VvLDOX* is responsible for encoding leucoanthocyanidin dioxygenase, a critical enzyme in the anthocyanin synthesis pathway responsible for the conversion of colorless anthocyanins to pigmented anthocyanins. All treatments notably sustained *VvLDOX* expression, with the CS and CS+Se groups showing the most significant effects (Figure 6B). This implied that chitosan treatment might play an important role in maintaining *VvLDOX* expression.

*VvUFGT* encodes UDP-glucose–flavonoid 3-O-glucosyltransferase. The expression of *VvUFGT* was observed to fluctuate across the storage period, notably in response to different treatments. The CK group displayed an initial increase, followed by a decrease in *VvUFGT* expression. On the 15th day, the CS treatment alleviated the decrease in *VvUFGT* expression, while both the Se and CS+Se groups exhibited lower *VvUFGT* expression than the CK group. However, by the 30th day, all groups showed a similar level of decreased expression (*p* > 0.05) (Figure 6C). This may indicate that chitosan treatment possibly influenced early-stage anthocyanin formation, while selenium accelerated the decrease in *VvUFGT* expression. The decrease in *VvUFGT* expression may be related to the regulation of the anthocyanin synthesis pathway and the accumulation of anthocyanin.

*VvOMT* encodes O-methyltransferase, primarily involved in modifying anthocyanin molecules through O-methylation. The expression of *VvOMT* continued to decline across all groups, almost halting after 30 days. The Se group demonstrated the most rapid decline, reflecting a gradual fading of the anthocyanin synthesis pathway during storage, thereby reducing the necessity for *VvOMT* expression. However, in the CK group, *VvOMT* expression rebounded after reaching a low point in the latter stages of storage. This rebound may be associated with complex physiological and metabolic changes triggered by decay in the CK group during late-stage storage, subsequently impacting gene expression (Figure 7A).

*Vv5GT* encodes UDP-glucose–flavonoid 5-O-glucosyltransferase primarily responsible for adding glucose moieties to anthocyanin molecules. Vv5GT expression was upregulated across all treatment groups during storage, with variations in timing and intensity of effects. The CS group predominantly influenced expression at the beginning of storage, while the Se group affected expression more towards the end of storage. In contrast, the CS+Se group demonstrated an intermediate effect, reaching the highest expression on the 30th day (Figure 7B). This suggested that different treatments exerted different temporal and quantitative influences on anthocyanin modification through glucosylation.

*VvGST*, encoding glutathione S-transferase and playing a crucial role in anthocyanin metabolism and distribution, experienced a sharp decline in expression across all groups and stabilized after the 30th day. However, in the CS+Se group, the expression rebounded by the 30th day, while the other groups exhibited a considerable reduction in expression (Figure 7C). This suggested a decreased demand for significant anthocyanin accumulation during the last storage stages, leading to a decrease in *VvGST* expression. Nevertheless, the expression remained relatively low to maintain the pericarp’s color.

### 3.5. Correlation Analysis

Figure 8 presents the results of the correlation analysis between the CK (A) and CS+Se (B) groups. In the CS+Se treatment, a strengthened correlation between total phenolic, flavonoid, and anthocyanin compared to the CK group was observed, resulting in a notably significant positive correlation (*p* < 0.01) between total phenolic content and flavonoid content and a significant positive correlation (*p* < 0.05) between anthocyanin content and flavonoid content. In the CK group, anthocyanin content exhibited a highly significant and positive correlation with *VvLDOX* expression (*p* < 0.01). Moreover, the CS+Se treatment weakened the correlation between anthocyanins and the expression of *VvDFR*, *VvLDOX*, *VvUFGT*, *VvOMT*, and *VvGST* when compared to the CK group. In the CK group, anthocyanins displayed a significant positive correlation with the expression of *VvCHS* and a significant negative correlation with the expression of *VvCHI*, whereas the CS+Se group showed an opposite trend compared to the CK group.

### 3.6. Cluster Analysis

Figure 9 illustrates a heat map derived from cluster analysis showing the expression patterns of flavonoid-pathway-related genes in each group across different storage periods. Our segmentation of grape samples from each group during various storage intervals into four categories allowed for a comprehensive examination. The fluctuations observed in *VvCHI*, *VvF3′5′H*, *VvCHS*, and *VvDFR* across different treatment groups during grape storage were particularly notable, suggesting that these treatments noticeably influenced the regulation of these four genes. Notably, the expression of *VvCHI* in groups I and IV surpassed that in groups II and III. The late storage period for the CK and Se groups was predominantly concentrated in groups I and IV, coinciding with the rotting phenomenon in these groups. This observation suggested a potential link between heightened *VvCHI* expression during the late storage period and the occurrence of rot. Moreover, *VvF3′5′H* expression in group IV was higher than that in other groups, with CS+Se representing this cluster, potentially indicating a marked upregulation of *VvF3′5′H* by the CS+Se treatment. The expression of *VvCHS* and *VvDFR* in group II exceeded that of other groups. This group encompassed all treatments except for the CK, indicating that Se, CS, and CS+Se treatments induced upregulation in the expression of *VvCHS* and *VvDFR*. Notably, among these, CS+Se treatment, specifically represented by CS+Se-60, displayed the most vivid coloration, indicating that CS+Se treatment had a more pronounced effect on the regulation of *VvCHS* and *VvDFR* during the later stage of grape storage.

## 4. Discussion

Investigations involving selenium–chitosan primarily focused on applications in animal and medical fields, with limited exploration in post-harvest applications for fruits and vegetables. In our previous study, we found that selenium–chitosan treatment was effective in maintaining storage quality and volatile components of grapes [35,36], and the present study aimed to expand upon existing studies and conduct a thorough investigation into the effect of selenium–chitosan treatment on the preservation of grape skin color during storage. The findings indicated that the selenium–chitosan treatment significantly outperformed the individual treatments of selenium or chitosan alone in maintaining the color of grape skins.

The color analysis showed that the CS+Se group significantly excelled in maintaining the grape skin color. This efficacy might be attributed to the interplay between selenium and chitosan, leveraging their antioxidant and structural properties. Selenium is recognized for its antioxidant properties, which contributes to inhibiting the oxidation of anthocyanin and other pigments. In addition, chitosan creates a protective barrier on the skin surface that minimizes color deterioration caused by water loss and external factors [40]. The combined effect of these dual treatments synergistically enhanced the preservation of grape pericarp color. This experimental outcome mirrors observations made in broccoli [37].

Quantitative analysis of total phenolic, flavonoid, and anthocyanin further confirmed the effectiveness of CS+Se treatment in maintaining grape skin color. The trends were observed as an initial increase followed by a gradual decrease, which is consistent with the changes observed in “Hutai” 8 grapes during storage at low temperatures [41]. The increase in these compound levels at the beginning of storage may stem from continued post-harvest physiological activities coupled with the low-temperature stress experienced by the grapes, contributing to the accumulation of total phenolic, flavonoid, and anthocyanin [42,43]. By the end of the storage period, the total phenolic and flavonoid contents were higher than their initial values in all treatment groups, likely due to the release of phenolic compounds from their complexes with other components such as proteins and carbohydrates [44]. Flavonoids, particularly anthocyanins, are essential in establishing and maintaining grape skin color, and the regulation of flavonoid biosynthesis involves several enzymes [42]. The accumulation of flavonoids improves plant tolerance to cold and freezing stress by preventing protein aggregation and mitigating stress-related damage caused by oxidative stress, electrolyte leakage, and photosynthetic inactivation [45]. Throughout storage, the CS+Se group exhibited significant higher total phenolic, flavonoid, and anthocyanin contents compared to other groups, potentially because both CS [46] and Se [47] induced oxidative stress and enhanced the activities of enzymes involved in flavonoid biosynthesis and the CS+Se treatment can superimpose the antioxidant effects of the two in a synergistic manner. Similar effects were observed with burdock oligofructose treatment of blueberries [48].

In comparison to the CK group, the CS+Se treatment intensified the correlation between total phenolic, flavonoid, and anthocyanin, suggesting that the CS+Se treatment may impact common regulatory mechanisms such as enzyme activity, gene expression, or metabolic pathways, enhancing their accumulation and reinforcing the relationship between total phenolic, flavonoid, and anthocyanin. Typically, total phenolic, flavonoid, and anthocyanin exhibit antioxidant activity, suggesting that CS+Se treatment triggers the antioxidant potential of grapes.

Gene expression analysis provided insights into the dynamic variation patterns of gene expression associated with anthocyanin biosynthesis. CHS represented the pivotal stage in the flavonoid pathway for anthocyanin synthesis, catalyzing the condensation of malonyl coenzyme A and 4-coumaroyl coenzyme A to form tetrahydroxychalcone [49,50]. Different treatment groups exhibited distinct expression profiles, and CS+Se treatment significantly upregulated *VvCHS* expression. This indicated that CS+Se treatment might have triggered the activation of the anthocyanin synthesis pathway. The synergistic effect of CS and Se was even more pronounced in the later storage stages. *VvCHI* is essential for anthocyanin synthesis and our findings suggested that its expression may be correlated with microbial infestation in grapes. Flavonoids are highly sensitive to environmental stress and storage conditions as part of their adaptative mechanisms [51]. When grapes are subjected to microbial attack, increased *VvCHI* expression might contribute to the synthesis of more flavonoids. Certain flavonoid components exhibit antimicrobial activities that assist in mitigating or counteracting potential damage caused by microbial attacks. Moreover, the plant’s immune system likely plays a role in this process. Microbial invasion triggers the plant’s immune response, leading to a series of changes in signaling and gene expression, and the regulation of *VvCHI* could be part of this immune response, enhancing the plant’s capacity to protect itself.

Flavanone 3-hydroxylase (F3H), flavonoid 3′-hydroxylase (F3′H), and flavonoid 3′,5′-hydroxylase (F3′5′H) are the key enzymes in the anthocyanin biosynthesis pathway responsible for synthesizing flavonoids. In grape metabolism, the flavonoid pathway does not efficiently use monohydroxylated intermediates; therefore, F3′H and F3′5′H are essential for producing dihydroxylated and trihydroxylated intermediates [51,52]. F3H initiates the 3-hydroxylation of flavanones, generating precursors of anthocyanidins. F3′H introduces 3′-hydroxylation to produce precursors of cyanidins, while F3′5′H catalyzes 3′,5′-hydroxylation to produce precursors of delphinidin under certain conditions. Variation in *VvF3H*, *VvF3′H*, and *VvF3′5′H* expression across different treatment groups affects anthocyanin composition [42,53,54]. Studies have demonstrated that a reduction in F3′H or F3′5′H activity can impact the composition of flavonoids, though not significantly altering the synthesis of flavonoids in this pathway [52]. Both the CS and CS+Se groups exhibited an upregulation in *VvF3′5′H* expression, suggesting that chitosan might have increased the proportion of the pathway leading to delphinidin. Consequently, grape skins from these groups potentially contain a higher proportion of delphinidin.

DFR, LDOX, and UFGT are important in the final stages of anthocyanin synthesis, where anthocyanidins transform into anthocyanin [53]. DFR serves as a crucial enzyme in the anthocyanin synthesis pathway, catalyzing the conversion of flavanones to flavonoids, thus affecting pericarp color shifts. Studies have shown that low-temperature exposure does not increase the expression levels of *VvDFR* and *VvUFGT*, maintaining negligible levels in samples stored at 4 °C and 25 °C [55]. At the early storage stage, the CK group exhibited higher *VvDFR* expression compared to other groups due to relatively vigorous metabolic activity, declining in the late storage stage due to reduced vitality caused by decay. Other groups, undergoing preservation treatments, displayed stable *VvDFR* expression due to sustained vitality. LDOX is the rate-limiting enzyme in anthocyanin synthesis and directly affects anthocyanin formation. Se seemed to hinder the expression of *VvLDOX*. The CS and CS+Se groups displayed more consistent *VvLDOX* expression, indicating that chitosan plays a predominant role in maintaining *VvLDOX* expression. Chitosan might affect *VvLDOX* expression through various pathways, such as regulating transcriptional activity by interacting with signaling pathways, post-transcriptional modifications like methylation, and optimizing LDOX function by stabilizing the intracellular environment. Anthocyanidins require stabilization through glycosylation of the hydroxyl group on the C ring to form the corresponding glycoside [56,57]. UFGT is a key enzyme responsible for adding glucose moieties to the anthocyanidin molecule. The experimental results indicated a continuous decrease in *VvUFGT* expression during storage, especially prominent in the CK group, which is consistent with previous findings [53]. This decrease might be related to the degree of anthocyanin accumulation. In the CS group, we observed that the decrease in *VvUFGT* expression was less pronounced early in storage, suggesting that chitosan treatments might actively influence anthocyanin formation. These treatments could potentially impact *VvUFGT* expression through different mechanisms, including transcriptional regulation via intracellular signaling pathways, affecting intracellular substrate availability, enzyme activity, and the metabolic state, influenced by storage temperature and treatments.

*VvOMT* encodes an O-methyltransferase responsible for O-methylation modification in anthocyanin molecules. The findings indicated a decrease in *VvOMT* expression across all treatment groups during storage, with nearly ceasing expression after the 30th day. This gradual fading in the anthocyanin synthesis pathway during storage might account for the reduced need for *VvOMT* expression. An interesting observation was the rebound in *VvOMT* expression within the CK group after reaching the lowest point during the late storage period. This resurgence might be attributed to the complex physiological and metabolic changes due to decay in the CK group during late-stage storage, thereby impacting gene expression. This phenomenon highlighted that the fruit’s physiological state in the late storage period could unpredictably affect gene expression. *Vv5GT* encodes the UDP-glucose–flavonoid 5-O-glucosyltransferase enzyme responsible for adding glucose moieties to anthocyanin molecules. The experimental data revealed an upsurge in *Vv5GT* expression in all treatment groups post-storage. CS treatment impacted the pre-storage period more, while Se treatment had a greater effect during the post-storage period. The CS+Se treatment displayed intermediate effects. This suggested that chitosan could impact anthocyanin modification by regulating *Vv5GT* expression during pre-storage, while selenium treatment might have further improved this regulatory effect during the late storage period. This phase-dependent regulation could be associated with the varying requirement for anthocyanin synthesis and modification during various storage stages. In terms of the upstream and downstream genes of *Vv5GT*, the expression of *Vv5GT* remained stable and did not significantly change during storage. *VvGST* encodes a glutathione S-transferase enzyme. In some cases, anthocyanins can be toxic, but GST facilitates the binding of anthocyanins to glutathione for excretion. These conjugates reduce the toxicity of anthocyanin ketones and enhance their stability. Intracellularly, these conjugates can be translocated to various cellular substructures, such as vesicles or other organelles, to store and distribute anthocyanins [58]. The experimental results indicated a rapid decrease in *VvGST* expression in all treatment groups of post-harvest grape. This suggested a lower accumulation of anthocyanin and indicated that their expression level remained relatively low, primarily to maintain skin color. The expression rebound in the CS+Se group on the 30th day suggested that this treatment promoted the accumulation of anthocyanins in the mid-storage period and helped to maintain the stability and storage distribution of anthocyanins in the late storage period. On the other hand, the CS+Se treatment weakened the correlation between anthocyanin and the expression of related genes in the *VvDFR* and its downstream related genes compared to CK. This result appears contradictory to the enhanced correlation between total phenolic, flavonoid, and anthocyanin. However, an analysis aligning the observed and experimental results suggested that the rate of anthocyanin synthesis is substantially slow in post-harvest storage, leading to a shortage of precursor materials. This insufficiency determined the reaction speed in the latter phase of the flavonoid pathway, influencing the rate of anthocyanin synthesis. The CS+Se treatment effectively addressed this limitation by bolstering the expression of flavonoid-pathway-related genes to ensure ample accumulation of anthocyanin precursors. This treatment mitigated the factor that solely dictated the rate of anthocyanin synthesis, leading to an overall enhanced expression in the latter half of the flavonoid pathway reaction.

We also examined the role of *VvLDOX* in the flavonoid pathway, identifying it as a critical factor affecting the rate of anthocyanin synthesis. Our investigation into the expression pattern of *VvLDOX*, *VvUFGT*, *VvOMT*, and *VvGST* indicated a consistent decreasing trend during storage, with the rate of decrease gradually lessening, eventually stabilizing. This trend suggested a form of negative feedback regulation, where the rate of anthocyanin synthesis became increasingly affected by substrate concentration as the flavonoid pathway progressed. Among these genes, *VvUFGT*, *VvOMT*, and *VvGST* did not exhibit significant differences between the CS+Se group and the CK group. However, a notable variation was observed in the expression of *VvLDOX* (*p* < 0.05). Hence, our findings confirmed that *VvLDOX* functions as the rate-limiting enzyme for anthocyanin synthesis during grape storage.

## 5. Conclusions

This study investigated the impacts of selenium–chitosan treatment on grape color preservation during storage. The findings highlighted that the selenium–chitosan treatment effectively reduced the decay rate, maintained the soluble solids, and preserved the natural color of the grapes. Moreover, this treatment significantly inhibited the browning process and curbed the decline in total phenolic, flavonoid, and anthocyanin during storage. These effects can be attributed to the antioxidant characteristics of selenium and the protective barrier formed by chitosan, effectively reducing pigment oxidation and minimizing water loss. Gene expression analysis revealed that the CS+Se treatment not only increased the expression of *VvCHS*, *VvF3H*, *VvF’3′H*, *VvLDOX*, and *Vv5GT* but also stabilized the expression of *VvCHI*, *VvF3′H*, and *VvDFR* throughout storage, thereby facilitating the accumulation and stabilization of anthocyanins. In summary, the selenium–chitosan treatment proved to be highly effective in preserving the color of grape during the storage process compared to CS or Se treatments alone.

## Figures and Tables

**Figure 1 foods-13-00499-f001:**
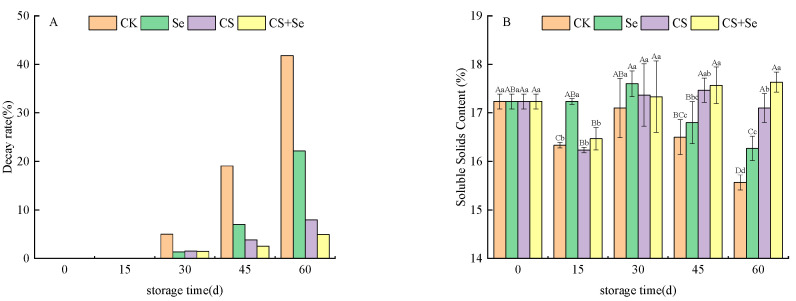
The decay rate (**A**) and soluble solids content (**B**) of “Red Globe” grapes during storage. CK, control; Se, sodium selenite (25 mg L^−1^); CS, chitosan (1.0%); CS+Se, selenium–chitosan (25 mg L^−1^ selenium and 1.0% chitosan). Bars indicate standard deviation (±SD). Different uppercase letters indicate significant differences between different storage times for the same group and different lowercase letters indicate significant differences between different groups for the same storage time (*p* < 0.05).

**Figure 2 foods-13-00499-f002:**
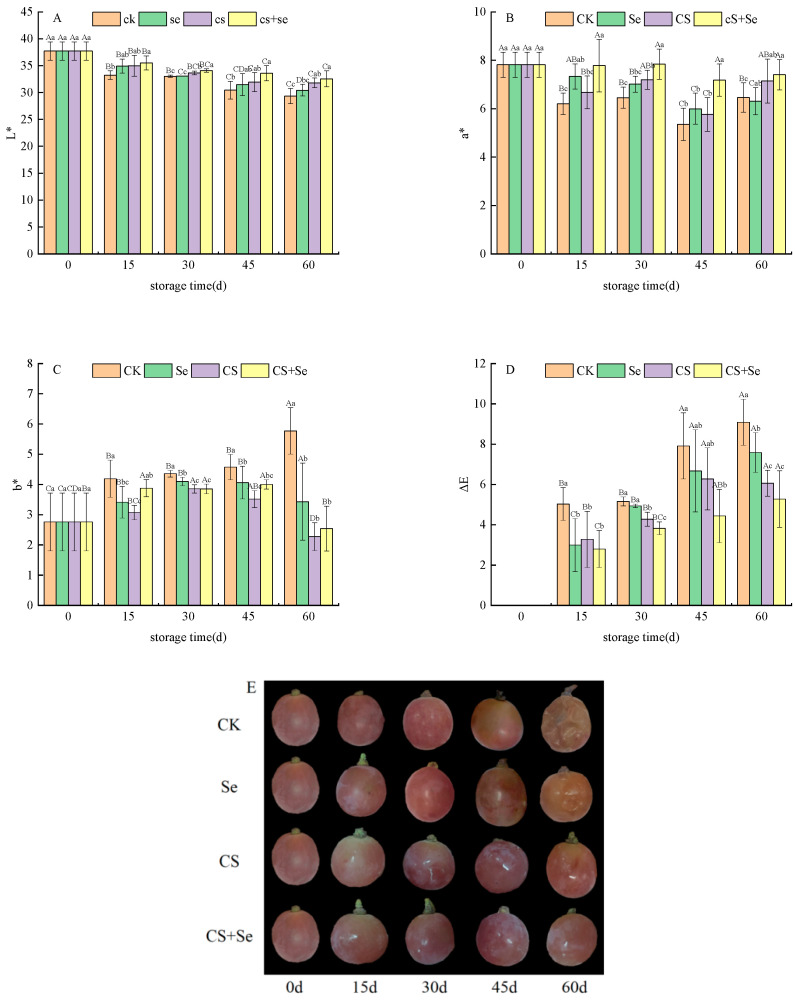
Chromatic aberration of “Red Globe” grapes during storage. L* (**A**). a* (**B**). b* (**C**). ΔE (**D**). Phenotype (**E**). CK, control; Se, sodium selenite (25 mg L^−1^); CS, chitosan (1.0%); CS+Se, selenium–chitosan (25 mg L^−1^ selenium and 1.0% chitosan). Bars indicate standard deviation (±SD). Different uppercase letters indicate significant differences between different storage times for the same group and different lowercase letters indicate significant differences between different groups for the same storage time (*p* < 0.05).

**Figure 3 foods-13-00499-f003:**
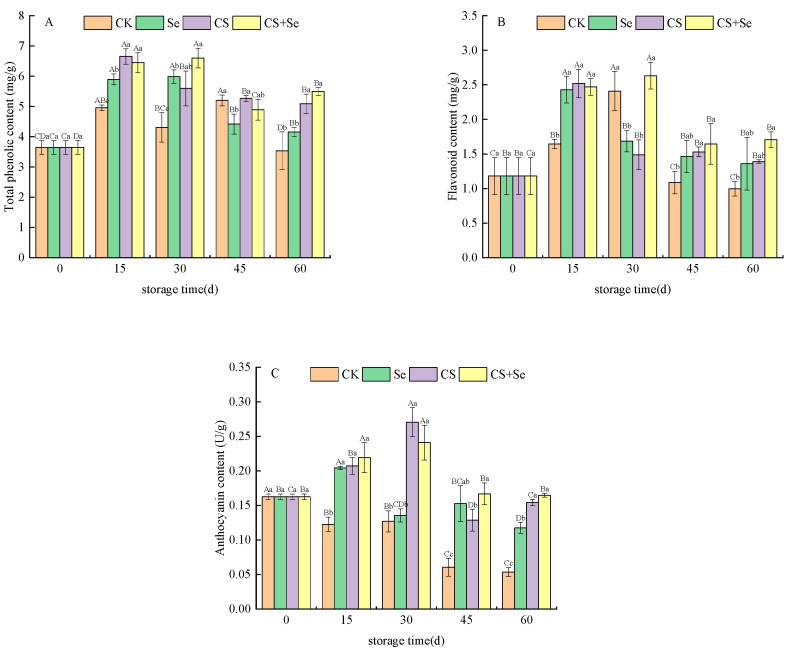
The content of total phenolic (**A**), flavonoid (**B**), and anthocyanin (**C**) of “Red Globe” grapes during storage. CK, control; Se, sodium selenite (25 mg L^−1^); CS, chitosan (1.0%); CS+Se, selenium–chitosan (25 mg L^−1^ selenium and 1.0% chitosan). Bars indicate standard deviation (±SD). Different uppercase letters indicate significant differences between different storage times for the same group and different lowercase letters indicate significant differences between different groups for the same storage time.

**Figure 4 foods-13-00499-f004:**
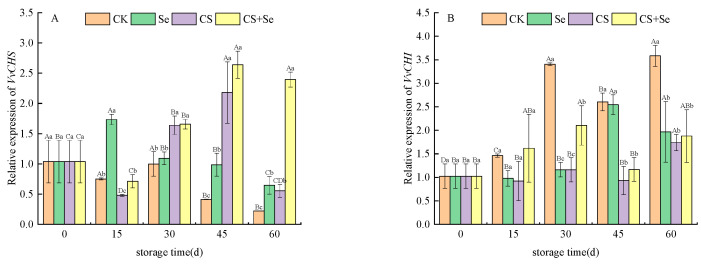
Relative expression levels of *VvCHS* (**A**) and *VvCHI* (**B**) of “Red Globe” grapes during storage. CK, control; Se, sodium selenite (25 mg L^−1^); CS, chitosan (1.0%); CS+Se, selenium–chitosan (25 mg L^−1^ selenium and 1.0% chitosan). Bars indicate standard deviation (±SD). Different uppercase letters indicate significant differences between different storage times for the same group and different lowercase letters indicate significant differences between different groups for the same storage time (*p* < 0.05).

**Figure 5 foods-13-00499-f005:**
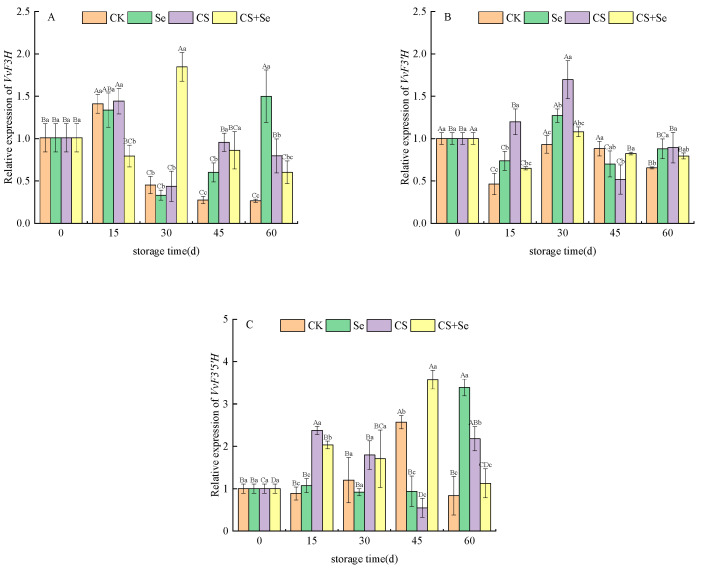
Relative expression of *VvF3H* (**A**), *VvF3′H* (**B**), and *VvF3′5′H* (**C**) of “Red Globe” grapes during storage. CK, control; Se, sodium selenite (25 mg L^−1^); CS, chitosan (1.0%); CS+Se, selenium-chitosan (25 mg L^−1^ selenium and 1.0% chitosan). Bars indicate standard deviation (±SD). Different uppercase letters indicate significant differences between different storage times for the same group and different lowercase letters indicate significant differences between different groups for the same storage time (*p* < 0.05).

**Figure 6 foods-13-00499-f006:**
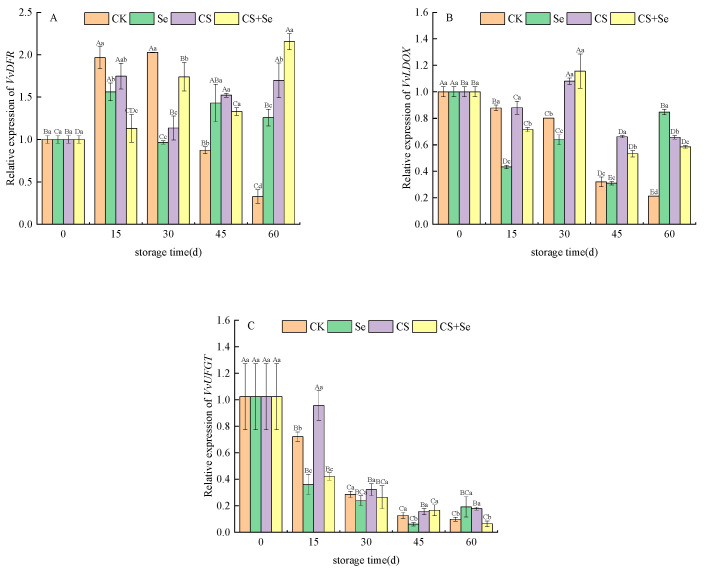
Relative expression of *VvDFR* (**A**), *VvLDOX* (**B**), and *VvUFGT* (**C**) of “Red Globe” grapes during storage. CK, control; Se, sodium selenite (25 mg L^−1^); CS, chitosan (1.0%); CS+Se, selenium–chitosan (25 mg L^−1^ selenium and 1.0% chitosan). Bars indicate standard deviation (±SD). Different uppercase letters indicate significant differences between different storage times for the same group and different lowercase letters indicate significant differences between different groups for the same storage time (*p* < 0.05).

**Figure 7 foods-13-00499-f007:**
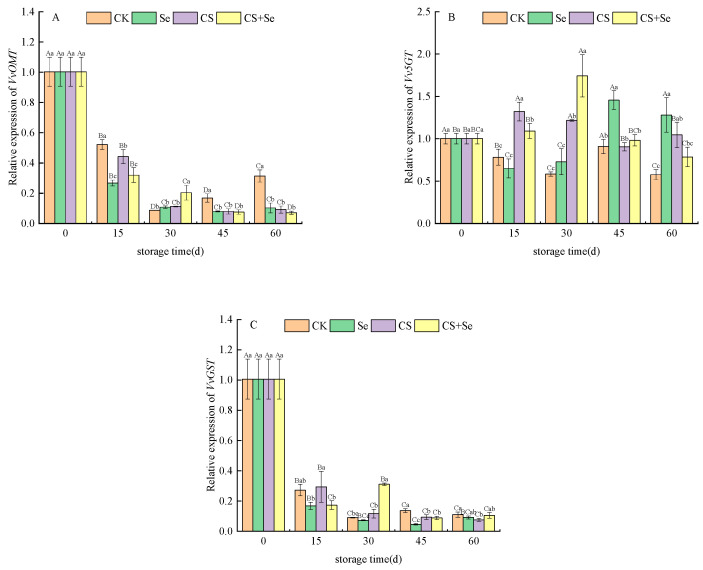
Relative expression of *VvOMT* (**A**), *Vv5GT* (**B**), and *VvGST* (**C**) of “Red Globe” grapes during storage. CK, control; Se, sodium selenite (25 mg L^−1^); CS, chitosan (1.0%); CS+Se, selenium-chitosan (25 mg L^−1^ selenium and 1.0% chitosan). Bars indicate standard deviation (±SD). Different uppercase letters indicate significant differences between different storage times for the same group and different lowercase letters indicate significant differences between different groups for the same storage time (*p* < 0.05).

**Figure 8 foods-13-00499-f008:**
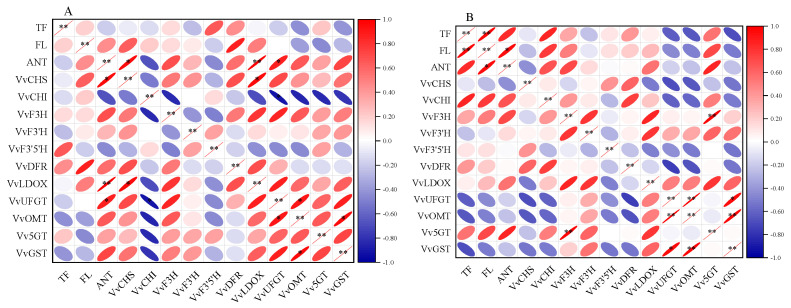
Heat map of correlation of each datum during storage of “Red Globe” grape in CK group (**A**) and CS+Se group (**B**). Red color indicates positive correlation and blue color indicates negative correlation. The flatter the oval shape, the greater the correlation. The * indicates significant differences at the 0.05 probability level and ** indicates highly significant differences at the 0.01 probability level.

**Figure 9 foods-13-00499-f009:**
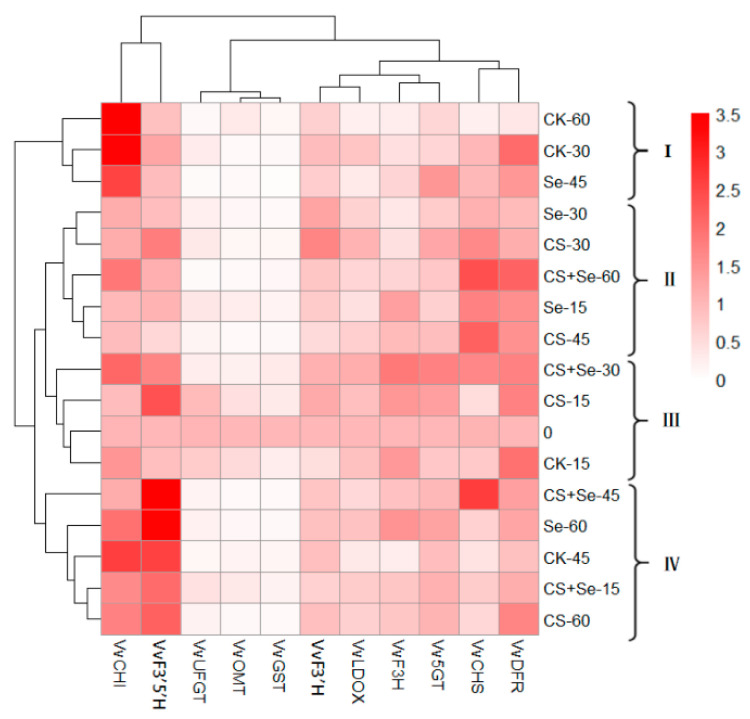
Heat map of cluster analysis illustrating the gene expression associated with the flavonoid pathway across different treatment groups of “Red Globe” grapes throughout various storage periods. The “Red Globe” grapes from multiple treatment groups over diverse storage time were classified into four distinct clusters.

**Table 1 foods-13-00499-t001:** Primers used for RT-qPCR.

Gene	Forward Primer Sequence (5′-3′)	Reverse Primer Sequence (5′-3′)
*KyActin1*	GATTCTGGTGATGGTGTGAGT	GACAATTTCCCGTTCAGCAGT
*VvCHS*	GAAGATGGGAATGGCTGCTG	AAGGCACAGGGACACAAAAG
*VvCHI*	CAGGCAACTCCATTCTTTTC	TTCTCTATGACTGCATTCCC
*VvF3H*	CCAATCATAGCAGACTGTCC	TCAGAGGATACACGGTTGCC
*VvF3′H*	GCCTCCGTTGCTGCTCAGTT	GAGAAGAGGTGGACGGAGCAA
*VvF3′5′H*	AAACCGCTCAGACCAAAACC	ACTAAGCCACAGGAAACTAA
*VvDFR*	GAAACCTGTAGATGGCAGGA	GGCCAAATCAAACTACCAGA
*VvLDOX*	AGGGAAGGGAAAACAAGTAG	ACTCTTTGGGGATTGACTGG
*VvUFGT*	GGGATGGTAATGGCTGTGG	ACATGGGTGGAGAGTGAGTT
*VvOMT*	GTTCAACTTCATGAGATGGA	GGAGAACTACCTCAACTACCA
*Vv5GT*	TTCCATGGCTGAACTCAC	AACATCCAACTGCTTGGTGAC
*VvGST*	AAAAGTCATGGAGCTCGCTG	CAGCTTCCTTCACCAAGTAT

## Data Availability

The data presented are contained within the article.

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
