# Peer review of "Impacts of Selenium–Chitosan Treatment on Color of “Red Globe” Grapes during Low-Temperature Storage"

_foods, 2024, doi:10.3390/foods13030499_

Round 1

Reviewer 1 Report

Comments and Suggestions for Authors

The manuscript has demonstrated that the selenium-chitosan treatments have been useful to preserve the skin color of the red grapes during storage. The manuscript is of interest to the scientific community because in this way browning of the skin is avoided, while preserving the bioactive compounds of interest and the expression of the genes. The work is well performed and organized. However, I list below some aspects and questions that have not been clear to me for further explanation, as well as some comments and opinions.

Lines 102-103: A study of the storage of grapes in this selenium-chitosan mixture has already been previously carried out. Therefore, what is the novelty and importance of carrying out this work? Emphasize the objective of the work and the main contribution it makes.

It would be very convenient to include a new section in the Materials and Methods indicating all the reagents or solvents used in the work, specifying the grade or type of reagent, as well as the company, city, and country where they were purchased. In addition, ensure all sources of equipment are included and compliant with the Guide for Authors, i.e. equipment model, company name, city and country.

Line 117: How was that selection made? Manually?

Section 2.5.: Was the grape weighed directly or pre-crushed in order to increase the contact surface? In addition, plant matrices usually have large amounts of water that can interfere with the extraction of the bioactive compounds of interest. Wouldn't it be convenient to subject the sample to a lyophilization process to completely remove that water?

Line 152: How was this agitation carried out? What equipment was used?

Line 156: It would be convenient to indicate which wavelength was used for each family of compounds.

In the results section, how many replicates were made of each measurement or analysis of the different parameters? It would be convenient to indicate it in the text.

Figure 1A: It would be convenient to include the error bars as well as the significant differences between each of the values.

Section 3.3: Significant differences refer to the four treatments within the same day or to the same treatment on different days of storage. This should be explained much better.

How do the different treatments carried out on the grapes influence the concentration of bioactive compounds?

Section 3.5: Why has a different level of significance been used (p-Value < 0.05 or 0.01) depending on the parameter analyzed?

Lines 406-407: It would be convenient to indicate here some of these references of research with selenium and chitosan in fruits and vegetables.

Additionally, you must take care of some formal aspects of the manuscript:

Title: Capitalize each word according to the format of the journal.

Lines 130, 135, 139, 144, 166,…: Capitalize each word according to the format of the journal. Unify and apply to the entire manuscript.

Put a separation after the number and before “ºC”.

Put a separation after and before “±”, and “=”. Unify and apply to the entire manuscript.

References 9, 53: The DOI is missing.

References: The name of the journal would be always in abbreviated format, and after each abbreviated word there should be a dot “.”.

References 7, 10, 17, 25, 43,…: Scientific names should be in italics. Unify and apply to the entire manuscript.

Comments on the Quality of English Language

English is fine, just review some expressions and formal aspects for final publication.

Reviewer 2 Report

Comments and Suggestions for Authors

Title needs to be improved; I suggest to change color of skin to color of grapes as you studied the color of grapes using coloremeter and mention the storage condition in title

Keyword: selenium and chitosan should be separated

Line 28- 30: I think it is a overstatement about the role of grapes in human life and importance of it as a fruit?

Line 102: Which one of the authors do you indicate here by referring “my”?

21. Plant material and method: Did the grapes removed from the stem or they were immersed in coating solution as a cluster attached to a stem?

What was the condition of drying coated grapes? Was it forced air? Duration of drying?

2.4 chromatic observation: I am wondering if you can state the color of skin or it is infact the colour of grapes as the transparency of grape skin the data recorded is the colour of the grape itself and not separated skins only? Moreover, more details about the colour measurement is needed like the number of samples were measured for each treatment

2.5 phenolics antioxidants, flavoids: what instrument used for the absorbance analysis?

Line 200-205: Please explain the reason of the observation for reducing and then increase in TSS for Cs+SE samples?

3.2 chromatic observation? Why did author didn’t use the browning index instead of ΔE? Browning index is more suitable and ΔE is for measuring the colour difference between samples

Figure 2: what does the letters on the bars indicate? The difference between treatments in same day or same treatment in different days? The letters and significant difference between samples should be re-checked. For example for the L* at the day 60 the difference between Se and CS+SE is insignificant (standard deviations overlaps). Check all graphs for the significant differences as I see the same mistake in all.

Figure 4.(A)  Standard deviations between CK and Se shows a significance difference at day 45. Check all the standard deviations and letters for the graphs. I see the same issue in the other graphs and letters are not matching with statistical analysis.
